# Antiarrhythmic Effect of Sacubitril-Valsartan: Cause or Consequence of Clinical Improvement?

**DOI:** 10.3390/jcm8060869

**Published:** 2019-06-18

**Authors:** António Valentim Gonçalves, Tiago Pereira-da-Silva, Ana Galrinho, Pedro Rio, Luísa Moura Branco, Rui Soares, Joana Feliciano, Rita Ilhão Moreira, Rui Cruz Ferreira

**Affiliations:** Department of Cardiology, Hospital de Santa Marta, Centro Hospitalar de Lisboa Central, 1169-024 Lisbon, Portugal; tiagopsilva@sapo.pt (T.P.-d.-S.); anaisabelgalrinho@gmail.com (A.G.); pedrosantosrio@gmail.com (P.R.); luisamourabranco@me.com (L.M.B.); ruimsoares3@gmail.com (R.S.); joanagomesfeliciano@gmail.com (J.F.); ritailhaomoreira@gmail.com (R.I.M.); cruzferreira@netcabo.pt (R.C.F.)

**Keywords:** Sacubitril-Valsartan, antiarrhythmic effect, QTc interval, QRS interval, mechanical dispersion index

## Abstract

Sacubitril/Valsartan (LCZ696) reduced sudden cardiac death in the PARADIGM-HF trial. However, the mechanism by which LCZ696 reduces ventricular arrhythmias remains unclear. The aim of this study was to compare electrocardiographic (ECG) parameters and mechanical dispersion index, assessed by left ventricular (LV) global longitudinal strain (GLS), before and after LCZ696 therapy. We prospectively evaluated chronic Heart Failure (HF) patients with LV ejection fraction ≤40%, despite optimal medical and device therapy, in which LCZ696 therapy was started, while no additional HF treatment was expected to change. ECG and transthoracic echocardiographic data were gathered in the week before starting LCZ696 and at six months of therapy. A semiautomated analysis of LV GLS was performed and mechanical dispersion index was defined as the standard deviation from 16 time intervals corresponding to each LV segment. Of the 42 patients, 35 completed the six month follow-up, since two patients died and five discontinued treatment for adverse events. QTc interval (451.9 vs. 426.0 ms, *p* < 0.001), QRS duration (125.1 vs. 120.8 ms, *p* = 0.033) and mechanical dispersion index (88.4 vs. 78.1 ms, *p* = 0.036) were significantly reduced at six months. LCZ696 therapy is associated with a reduction in QTc interval, QRS duration and mechanical dispersion index as assessed by LV GLS.

## 1. Introduction

The PARADIGM-HF trial showed that Sacubitril/Valsartan (LCZ696), with the combination of neprilysin inhibitor and angiotensin II receptor blocker (ARB) could reduce both heart failure (HF) hospitalization and cardiovascular mortality by 20% in comparison with Enalapril [1]. As a result, LCZ696 has a Class I recommendation as a replacement for angiotensin-converting enzyme inhibitors (ACEI) for ambulatory patients with HF with reduced ejection fraction who remain symptomatic despite optimal treatment with an ACEI (or an angiotensin II receptor blocker (ARB), as an alternative, if they were not tolerant to ACEI), a beta-blocker (BB) and a mineralocorticoid receptor antagonist (MRA) [2].

A subanalysis of the PARADIGM-HF trial also showed a reduction in sudden cardiac death by 20% in relation to Enalapril, which does not differ amongst patients with or without an implantable cardioverter defibrillator (ICD) [3]. This antiarrhythmic effect was confirmed in two other studies, in which LCZ696 therapy was associated with a significant reduction in episodes of non-sustained and sustained ventricular tachycardia, appropriate ICD shocks, premature ventricular contractions and consequently, an increase in biventricular pacing percentage [4,5].

This antiarrhythmic effect of LCZ696 therapy can be one of the advantages of the combination of neprilysin inhibitor and ARB in relation to ACEI, since ACEI does not seem to significantly reduce sudden cardiac death in HF patients [6,7,8]. However, the precise mechanism by which LCZ696 causes a decrease in ventricular arrhythmias remains unclear.

Several potential mechanisms have been linked to this antiarrhythmic effect. Whether this reduction is caused by reverse remodeling, the reduction in myocardial fibrosis, wall stretch or sympathetic nervous system activation is not fully understood [3,9,10].

Electrocardiographic (ECG) changes could add information for the protective mechanisms associated with LCZ696 therapy. One previous study showed a significant reduction in corrected QT interval (QTc) and in the interval from T-wave peak to T-wave end interval after 1 month of LCZ696 therapy [11]. However, this study had a retrospective design and a limited follow-up time.

Mechanical dispersion as assessed by strain echocardiography has been linked to the prediction of arrhythmic events in the HF population [12,13]. To the best of our knowledge, there is no published data regarding the effects of LCZ696 therapy on mechanical dispersion index, which could potentially add information for the understanding of the antiarrhythmic mechanisms associated with LCZ696 therapy.

The aim of this study was to prospectively compare ECG data and mechanical dispersion, as assessed by strain echocardiography, before and after LCZ696 therapy, in a real-world cohort of patients with chronic HF with reduced ejection fraction and optimized standard of care therapy.

## 2. Experimental Section

The investigation conforms to the principles outlined in the Declaration of Helsinki. The institutional ethics committee and the national committee for patient information protection (CNPD authorization number 5962) approved the study protocol. All patients provided written informed consent.

All authors made the decision to submit the manuscript for publication and assume responsibility for the accuracy and completeness of the analyses.

### 2.1. Patient Population

The study included a prospective single center analysis, in which patients were included from October 2017 to June 2018.

During this period, all ambulatory patients with left ventricular (LV) ejection fraction ≤40% under optimized standard of care therapy and New York Heart Association (NYHA) class ≥ II, were proposed to start LCZ696 therapy according to the current Guidelines [2].

### 2.2. Definition of Chronic HF with Optimized Standard of Care Therapy

Optimized standard of care therapy for chronic HF was defined as: (1) treatment for at least six months with maximum tolerated doses of an ACEI (or ARB if appropriate), a BB and a MRA. (2) ICD and/or cardiac resynchronization therapy (CRT) if indicated by the current guidelines. (3) The subject had been adequately treated per applicable standards for coronary artery disease and mitral regurgitation [2]. (4) No changes of treatment were expected for the next six months. 

### 2.3. Study Protocol

All patients provided written informed consent. Thereafter, clinical, laboratorial, ECG and transthoracic echocardiography data were obtained in the week before starting LCZ696 therapy.

A washout period of 36 h was used to allow switching from an ACEI to LCZ696. LCZ696 therapy was preferentially started at 49/51 mg twice daily or 24/26 mg twice daily in patients with a dose <10 mg/day of Enalapril or equivalent. The aim was to double the dose every 2 to 4 weeks to reach the target maintenance dose of 97/103 mg twice daily except in patients with systolic blood pressure <100 mmHg, symptomatic hypotension, hyperkalaemia >5.5 mEq/L or a decrease in glomerular filtration rate to less than 60 mL/min as assessed by the Cockcroft–Gault equation.

All patients were followed-up with for six months after LCZ696 initiation. Clinical assessment, laboratorial tests, ECG and transthoracic echocardiography were repeated after the six months of LCZ696 therapy.

Information regarding all the collected data is presented in Appendix A.

### 2.4. ECG

A standard 12 lead ECG, consisting of three limb leads (I, II and III), three augmented limb leads (aVR, aVL and aVF) and six precordial leads (V_1_–V_6_) were obtained at rest, before starting LCZ696 and at six months of therapy [14]. All the ECG measurements were performed manually by a cardiologist blinded to the patient data.

Resting heart rate was measured from the ECG. All the intervals were measured in milliseconds. PQ interval was defined from the beginning of the P wave until the beginning of the QRS complex. The QRS duration was measured from the onset to the end of the QRS waves [15]. The QT interval was determined from the start of the Q wave to the end of the T wave [16] and was corrected (QTc) for the heart rate [17]. The Sokolow–Lyon criteria was calculated to estimate LV hypertrophy (SV_1_ + RV_5/6_) [18].

### 2.5. Transthoracic Echocardiogram

A complete transthoracic echocardiography study was performed using the GE Vivid E95 ultrasound system, with a frame rate of more than 60frames/sec, before starting LCZ696 and at six months of therapy.

LV ejection fraction was calculated by the biplane Simpson′s method of discs.

To calculate LV mechanical dispersion, a semiautomated analysis of speckle-based strain was performed after two-dimensional images were acquired in the standard apical four-, three- and two-chamber views. The time interval from the peak R-wave to peak negative strain was assessed in each LV segment. Mechanical dispersion index was defined as the standard deviation from these 16 time intervals [11].

### 2.6. Statistical Analysis

Baseline characteristics were summarized as frequencies (percentages) for categorical variables and as means and standard deviations for continuous variables. All analyses compare patients´ data at the baseline and after six months of LCZ696 therapy.

Normal distribution of continuous variables was verified by the Kolmogorov–Smirnov test. Categorical data were compared with Pearson’s *χ*^2^ test. The paired samples t-Test was used for the comparison of the variables before and after LCZ696 therapy. Statistical differences with a *p* value <0.05 were considered significant. Data was analyzed using the software Statistical Package for the Social Sciences for Windows, version 24.0 (SPSS Inc, Chicago, IL, USA).

## 3. Results

### 3.1. Overview of the Study Population

A total of 42 patients were enrolled in the study. Of the 42 patients, 35 (83.3%) completed the six month follow-up with LCZ696, since two (4.8%) patients died (one patient with intracranial hemorrhage after trauma not due to syncope and one patient with sudden cardiac death), and five (11.9%) patients discontinued treatment due to adverse events (two patients with reversible acute kidney injury and three patients with symptomatic hypotension with the lowest LCZ696 dose). No patient was lost during follow-up.

The baseline characteristics of the 35 patients who completed the six-month follow-up period with LCZ696 are presented in Table 1. Mean age was 58.6 ± 11.1 years, with 29 (82.9%) male patients and an ischemic etiology in 15 (42.9%) patients.

These patients were highly symptomatic, as revealed by a NYHA class ≥III in 51.4% and by 42.9% of hospitalizations for worsening HF in the year prior to LCZ696 therapy. All patients were previously using an ACEI or ARB and a beta-blocker. Furthermore, 94.3% were using an MRA. An ICD was previously implanted in 30 (85.6%) patients, out of which seven (20.0%) had a cardiac resynchronization therapy (CRT-D) system.

Sustained ventricular arrhythmias were documented in six (17.1%) patients in the six months prior to LCZ696 therapy. During the same LCZ696 therapy period, only two (5.7%) patients had a sustained ventricular arrhythmia. Of note, there was a reduction in the number of episodes of sustained ventricular arrhythmias, as assessed by the ICD monitoring, in these two patients. One patient had five episodes of sustained ventricular tachycardia in the monitoring zone plus 183 episodes of non-sustained ventricular tachycardia in the six months prior to LCZ696 therapy. In the same LCZ696 therapy period, no sustained ventricular tachycardia episodes were recorded, and there was a reduction to 70 episodes of non-sustained ventricular tachycardia. In the other patient, there was a reduction from five episodes of sustained ventricular tachycardia treated with burst pacing, to only two episodes of sustained ventricular tachycardia in the monitoring zone. The patient with sudden cardiac death was hospitalized at that time, with a documented third-degree atrioventricular block in cardiac telemetry, followed by cardiac arrest.

### 3.2. LCZ696 dose

LCZ696 therapy was started at 24/26 mg twice a day in 18 (51.4%) patients and 49/51 mg twice a day in 17 (48.6%) patients. At six months the dose was 24/26 mg twice a day in 10 (28.6%) patients, 49/51 mg twice a day in 11 (31.4%) patients and 97/103 mg twice a day in 14 (40.0%) patients.

There were no significant changes regarding the dose expressed as percent of the target dose of BB (68.8 ± 28.6% vs. 70.6 ± 28.0%, *p* = 0.278) and MRA (51.6 ± 19.0% vs. 53.2 ± 24.4%, *p* = 0.352) nor to the loop diuretic dose at the baseline and after six months of LCZ696 therapy.

Laboratory analysis showed no differences between the baseline and after therapy values of potassium (4.5 ± 0.4 vs. 4.6 ± 0.4 mEq/L, *p* = 0.292).

### 3.3. ECG Analysis

In the initial evaluation, 14 (40.0%) patients had a history of atrial fibrillation (AF): nine (25.7%) patients had permanent AF and five (14.3%) had paroxysmal AF. At six months, there were no additional patients in AF and none underwent catheter ablation. Table 2 presents the results of the collected ECG data.

In the 26 patients at sinus rhythm, no significant differences were found in PQ interval after LCZ696 therapy.

QRS duration and QTc interval were significantly reduced by 3.4% and 5.7%, respectively. A significant reduction in the SV_1_ + RV_5/6_ was also observed with LCZ696 therapy.

There were only seven (20%) patients with a CRT system, which did not allow the demonstration of significant differences regarding biventricular pacing percentage.

### 3.4. Transthoracic Echocardiogram Analysis

Table 3 presents the results of transthoracic echocardiogram analysis. LV dimensions and atrial volumes were significantly lowered after six months of treatment. LV ejection fraction (29 ± 6% vs. 35 ± 9%, *p* = 0.001) and global longitudinal strain (GLS) (−7.0 ± 2.6% vs. −8.9 ± 2.8%, *p* = 0.001) showed a significant improvement during the follow-up.

Mechanical dispersion was reduced by 10 ms (88 ± 28 ms vs. 78 ± 26 ms) with LCZ696 therapy. There were only eight (22.9%) patients with an increase in mechanical dispersion during follow-up, most of them (62.5%) also showing an increase in QTc interval.

## 4. Discussion

The antiarrhythmic effect of LCZ696 therapy is one of the clinical benefits of the combination of neprilysin inhibitor and ARB, compared to ACEI, since unlike BB [19,20] and MRA [21,22], ACEI does not seem to significantly reduce sudden cardiac death in HF patients [6,7,8].

Several potential mechanisms have been linked to this antiarrhythmic effect. One previous retrospective analysis of real-world patients revealed that a higher degree of reverse remodeling was associated with a lower burden of ventricular arrhythmias, as assessed by ICD monitoring [4]. Myocardial fibrosis, wall stretch or sympathetic nervous system modulation are other possible explanations [3,9,10]. The analysis of ECG changes could be one way to identify the electrical effects associated with LCZ696 therapy. One previous retrospective study showed a significant reduction in QTc interval (415 ± 20 ms vs. 409 ± 21 ms, *p* = 0.022) after one month of LCZ696 therapy.

To the best of our knowledge, this was the first prospective study evaluating the ECG changes with LCZ696 therapy. This was a group of chronic HF patients, with optimized standard of care therapy, and had a numerically higher percentage of patients treated with a baseline of BB (100% vs. 93.1%), MRA (94.3% vs. 52.2%), ICD (85.6% vs. 14.9%) and CRT (20% vs. 7%) when compared to the population studied in the PARADIGM-HF trial [1]. Nevertheless, our sample included highly symptomatic patients, as revealed by a NYHA class ≥III in 51.4% (only 23.9% in the PARADIGM-HF trial), 42.9% hospitalizations for worsening HF in the previous year and 17.1% of patients with ventricular arrhythmias in the six months prior to LCZ696 therapy.

LCZ696 therapy was started at 24/26 mg twice a day in 18 (51.4%) patients and 49/51 mg twice a day in 17 (48.6%) patients. This is in line with a recent World-Data study that started LCZ696 therapy at 24/26 mg twice a day in 51%, 49/51 mg twice a day in 38% and 97/103 mg twice a day in 11% of patients [23]. The daily dose of our trial at six months was slightly higher than the previous trial (251 mg/day vs. 207 mg/day) but lower than in PARADIGM-HF trial (375 mg/day) [1].

Despite no differences in the dose of BB, loop diuretic, amiodarone, digoxine or ivabradine, neither in the blood values of potassium at six months of treatment, the QRS duration and QTc interval were significantly reduced by 3.4% and 5.7%, respectively with LCZ696 therapy. A significant reduction was also observed in the SV_1_ + RV_5/6_. QTc interval and SV_1_ + RV_5/6_ reduction was significantly presented at six months of therapy irrespective of the starting LCZ696 dose.

One possible explanation for these finding is that QRS and QTc duration reductions are a consequence of the reverse remodeling process, since transthoracic echocardiography demonstrated a significant reduction in LV dimensions and atrial volumes after six months of LCZ696, as well as an improvement in LV ejection fraction (29 ± 6% vs. 35 ± 9%, *p* = 0.001) and GLS (−7.0 ± 2.6 vs. −8.9 ± 2.8, *p* = 0.001). However, ACEI therapy is also capable of promoting a reverse remodeling process and has not been linked to a reduction of sudden cardiac death, which suggests that the reverse remodeling process may not fully account for the antiarrhythmic effects of LCZ696 therapy. The inhibition of degradation of a number of endogenous vasoactive peptides by neprilysin, may explain additional antiarrhythmic effects, when compared to ACEI therapy [3,10].

Despite this we found no reduction in interventricular septum thickness at six months of LCZ696 therapy, previous studies had already shown that an increased LV mass will not change the QRS amplitude unless sufficient concurrent chamber dilatation was present [24,25]. Since our results revealed signs of reverse remodeling, as well as an improvement in LV ejection fraction, a reduction in the SV_1_ + RV_5/6_ could be expected.

Mechanical dispersion as assessed by strain echocardiography has been linked to the prediction of arrhythmic events in the HF population [12,13]. To the best of our knowledge, this was the first study evaluating the effects of LCZ696 therapy on mechanical dispersion index, with an absolute reduction of 10 ms (88 ± 28 vs. 78 ± 26 ms) with LCZ696 therapy. Figure 1 and Figure 2 show a case of mechanical dispersion index and GLS improvement in a patient with an ischemic dilated cardiomyopathy.

LV ejection fraction improvement, translated into a reduction from 32 (91.4%) patients with LV ejection fraction ≤35% at the baseline to only 14 (40%) patients after LCZ696 therapy. These findings, in association with a reduction of sustained ventricular arrhythmias, and the consistent reduction of electrical and mechanical dispersion, which reinforce the antiarrhythmic effects LCZ696, may suggest that ICD placement might be postponed in some patients starting LCZ696. This hypothesis should be the scope of future research, especially in patients with no ischemic etiology [26].

### Study Limitations

This study has some limitations. It is a single-center prospective experience and therefore the results might reflect local practice. Although the sample was not large, we were able to show a significant reduction in QTc interval and mechanical dispersion index in ECG and transthoracic echocardiography, respectively, that can help to explain some of the antiarrhythmic effects associated with LCZ696 therapy.

Despite being a prospective study comparing data at the baseline and after six months of LCZ696 therapy, there was no control group under ACEI or ARB therapy. However, after the results of the PARADIGM-HF trial [1], it would not be ethical to restrain patients from starting a therapy that has been shown to improve survival. In addition, we only included patients under optimized standard of care therapy (except for LCZ696 therapy) for more than six months, in whom no major changes in therapy were expected, including the initiation of new drugs, changes in dose regimens, electrical device implantations or other cardiac invasive procedures such as coronary revascularization, valvular treatment or catheter ablation of AF. These inclusion criteria were aimed at reducing bias and strengthening the causal relationship between LCZ696 initiation and changes in electrical and mechanical dispersion. Indeed, no differences were found in BB and MRA dosage after six months of therapy and there were no major invasive cardiac procedures during follow-up.

## 5. Conclusions

LCZ696 has been linked with an antiarrhythmic effect which is not fully understood. To the best of our knowledge, this was the first prospective study evaluating the ECG changes and the impact of LCZ696 on mechanical dispersion index as assessed by transthoracic echocardiography.

LCZ696 was associated with a reduction in QTc interval and QRS duration, as well as a reduction of mechanical dispersion index. To the best of our knowledge, this was the first prospective study evaluating the effects of LCZ696 therapy in electrical and mechanical dispersion. Our results may provide insights into the mechanisms underlying the reduction of sudden cardiac death with LCZ696.

Further trials should evaluate whether ICD implantation could be postponed in some HF patients treated with LCZ696 and no previous ventricular arrhythmias.

## Figures and Tables

**Figure 1 jcm-08-00869-f001:**
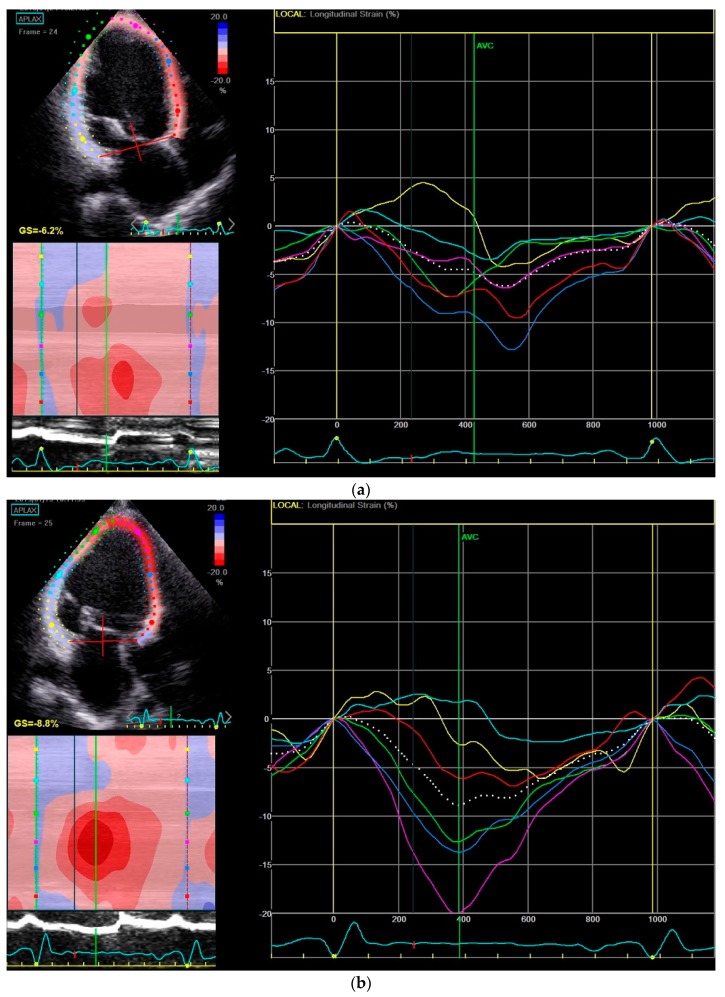
Mechanical dispersion index reduction before (**a**) and after (**b**) six months of Sacubitril-Valsartan (LCZ696) therapy in a patient with an ischemic cardiomyopathy in the APLAX view.

**Figure 2 jcm-08-00869-f002:**
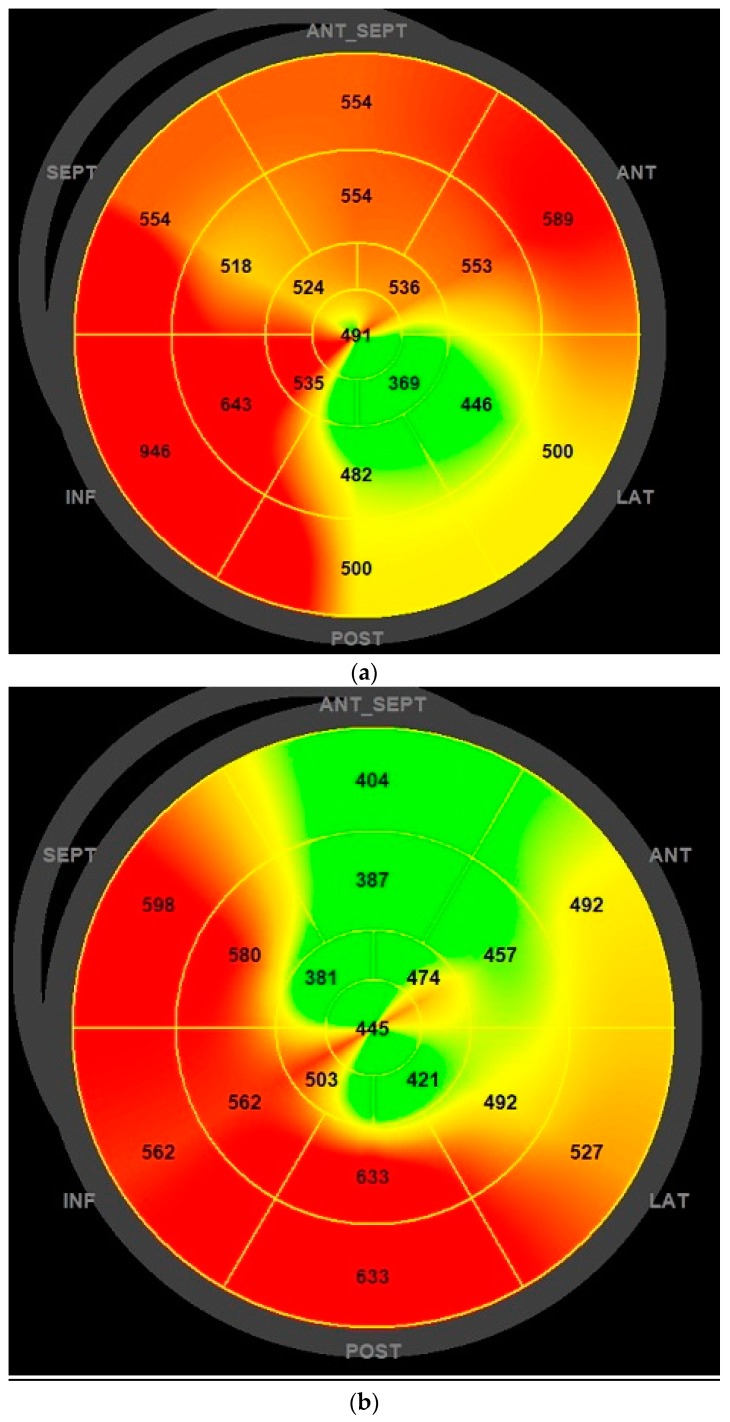
Mechanical dispersion bullseye plot reduction before (**a**) and after (**b**) six months of LCZ696 therapy.

**Table 1 jcm-08-00869-t001:** Baseline characteristics of the study population (*n* = 35).

Characteristics	*n* (%)
Mean age (years)	58.6 ± 11.1
Ischemic etiology	15 (42.9%)
Male gender	29 (82.9%)
New York Heart Association (NYHA) ≥ III	18 (51.4%)
Mean body mass index (kg/m^2^)	28.09 ± 3.77
Heart failure hospitalization in the previous year	15 (42.9%)
Median brain natriuretic peptide (BNP) (pg/mL) and interquartile range	314 (132 to 401)
Current smokers	7 (20.0%)
Previous hypertension	25 (71.4%)
Dyslipidemia	25 (71.4%)
Diabetes mellitus	11 (31.4%)
Peripheral artery disease	4 (11.4%)
Familiar history of heart failure	1 (2.9%)
Atrial fibrillation	14 (40%)
Chronic kidney disease	2 (5.7%)
Chronic liver disease	0 (0.0%)
Angiotensin-converting enzyme inhibitors	29 (82.9%)
Angiotensin II receptor blocker	6 (17.1%)
Beta-blockers	35 (100.0%)
Mineralocorticoid receptor antagonist	33 (94.3%)
Ivabradine	13 (37.1%)
Digoxin	9 (25.7%)
Amiodarone	9 (25.7%)
Implantable cardioverter defibrillator	30 (85.6%)
Cardiac resynchronization therapy (CRT-D)	7 (20%)
Percutaneous mitral-valve repair	3 (8.6%)

**Table 2 jcm-08-00869-t002:** Electrocardiographic data before and after six months of Sacubitril/Valsartan (LCZ696) therapy.

Electrocardiographic Data	Time 0	6 Months	*p*
Heart rate (bpm)	72.3 ± 13.0	67.1 ± 11.6	0.067
PQ interval (ms)	176.6 ± 21.4	174.6 ± 24.8	0.724
QRS duration (ms)	125.1 ± 33.5	120.8 ± 31.1	0.033
QTc interval (ms)	451.9 ± 48.1	426.0 ± 46.1	<0.001
SV_1_ + RV_5/6_ (mm)	21.2 ± 11.9	16.9 ± 9.8	0.001
Biventricular pacing (% *n* = 8)	97.4 ± 3.4	99.0 ± 0.8	0.183

Values are mean ± standard deviation.

**Table 3 jcm-08-00869-t003:** Echocardiographic data before and after six months of LCZ696 therapy.

Echocardiographic Data	Time 0	6 Months	*p*
Left ventricular end-diastolic diameter (mm)	71.3 ± 8.4	66.9 ± 7.6	0.001
Left ventricular end-systolic diameter (mm)	57.8 ± 9.4	53.1 ± 9.3	0.002
Interventricular septum thickness (mm)	9.6 ± 1.7	9.9 ± 1.9	0.280
Left ventricular ejection fraction (%)	29.3 ± 6.4	35.2 ± 8.6	0.001
Global longitudinal strain (%)	−7.0 ± 2.6	−8.9 ± 2.8	0.001
Mechanical dispersion (ms)	88.4 ± 28.1	78.1 ± 26.1	0.036
Left atrium volume (ml/m^2^)	51.5 ± 22.6	43.7 ± 15.8	0.004
Right atrium volume (ml/m^2^)	33.1 ± 4.4	28.5 ± 13.5	0.036
Tricuspid annular systolic excursion (mm)	19.2 ± 4.4	20.0 ± 4.8	0.404

Values are mean ± standard deviation.

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
