# Peer review of "Antiarrhythmic Effect of Sacubitril-Valsartan: Cause or Consequence of Clinical Improvement?"

_jcm, 2019, doi:10.3390/jcm8060869_

Round 1
Reviewer 1 Report
This manuscript by Gonçalves et al. investigated the antiarrhythmic effects of sacubitril-valsartan by prospectively evaluating the ECG parameters and left ventricular global longitudinal strain in a cohort of 42 heart failure patients before and 6 months after initiation of LCZ696 therapy. I have several minor comments:
• The starting dose of LCZ696 was different among patients. Did the authors note any difference in the significance of ECG and LV GLS parameters between those started with 49/51mg twice daily and those with 24/26mg twice daily?
• The authors write ‘…even in these 2 patients there were a reduction in the number of ventricular arrhythmias as assessed by the ICD monitoring.’ Please provide more detailed information on the number and type of arrhythmic episodes in these cases.
• Were the ECG parameters measured manually?
• The English needs to be edited. Several sentences are unclear. One example is the following: ‘All patients 136 were previously doing an ACEI or ARB and a beta-blocker and 94.3% were doing an MRA.’
• Many sections in the discussion are identical to the sections in the Introduction (see manuscript page 6, lines 184-199). The authors should refrain from copy-pasting the same sentences, and should rather focus on the implications of their findings in the context of previous reports. The whole discussion should be restructured, rewritten and shortened.
• The authors shall also discuss the significant changes observed in the ‘SV1 + RV5/6‘. Do the authors that this is due to a reverse remodeling of the LV within 6 months?
Author Response
Lisbon, 30th May 2019,
We are grateful for the insightful comments and suggestions and we have modified the manuscript in order to comply with all of them. We provide an itemized response to each of the points raised.
To: Reviewer 1
1- The starting dose of LCZ696 was different among patients. Did the authors note any difference in the significance of ECG and LV GLS parameters between those started with 49/51mg twice daily and those with 24/26mg twice daily?
LCZ696 therapy was started 24/26mg twice a day in 18 (51.4%) patients and 49/51mg twice a day in 17 (48.6%) patients.
At 6 months, the 18 patients who started with 24/26mg twice a day were doing the following LCZ696 dose:
· 24/26mg twice a day: 10 (55.6%) patients
· 49/51mg twice a day: 6 (33.3%) patients
· 97/103mg twice a day: 2 (11.1%) patients
The 17 patients who started with 49/51mg twice a day were doing the following LCZ696 dose:
· 24/26mg twice a day: 0 (0%) patients
· 49/51mg twice a day: 5 (29.4%) patients
· 97/103mg twice a day: 12 (70.6%) patients
A significant improvement at 6 months of LCZ696 therapy was present regarding GLS values in those who started with the lower (-6.6 ± 2.7% at baseline vs -8.6 ± 2.6% at 6 months, p=0.020) and with the medium (-7.4 ± 2.4% at baseline vs -9.3 ± 3.0% at 6 months, p=0.011) LCZ696 dose.
Despite a numerical reduction on mechanical dispersion values in the lowest starting dose group (85 ± 24 msec at baseline vs 81 ± 30 msec at 6 months, p=0.556), only those who started with the medium dose showed a significant reduction (92 ± 32 msec at baseline vs 75 ± 21 msec at 6 months, p=0.022). However, the reverse remodelling process (as assessed by left ventricular ejection fraction and GLS improvement), occurred in both groups. The absence of significant changes of mechanical dispersion in patients who started with the lowest LCZ696 dose could be explained by the lower initial values (85 ± 24 msec in low dose vs 92 ± 32 msec in medium dose) and the small size of this subgroup (18 patients).
Regarding ECG parameters, a significant reduction in QTc interval and SV1+RV5/6 occurred in both groups of patients. The same was not true for QRS duration reduction, since there was no statistical difference in patients who started with the lowest LCZ696 dose (133 ± 34 msec at baseline vs 129 ± 34 msec at 6 months, p=0.069) and with the medium LCZ696 dose (118 ± 33msec at baseline vs 114 ± 28 msec at 6 months, p=0.193). This could be explained by a lower magnitude of reduction in the QRS interval in comparison with the QTc interval and SV1+RV5/6 changes.
The summary of these results were added to the discussion section of the manuscript.
Considering the final dose of LCZ696 there were no significant interactions between the LCZ696 dose and the reduction in mechanical dispersion, QTc interval and SV1+RV5/6 changes.
2- The authors write ‘…even in these 2 patients there were a reduction in the number of ventricular arrhythmias as assessed by the ICD monitoring.’ Please provide more detailed information on the number and type of arrhythmic episodes in these cases.
In these two patients, ICD monitoring revealed a reduction in the number of ventricular arrhythmias. One patient had 5 episodes of sustained ventricular tachycardia in the monitoring zone plus 183 episodes of non-sustained ventricular tachycardia, in the 6 months previous to LCZ696 therapy. In the same LCZ696 therapy period, no sustained ventricular tachycardia episodes were recorded, and there was a reduction to 70 episodes of non-sustained ventricular tachycardia. In the other patient, there was a reduction from 5 episodes of sustained ventricular tachycardia to only 2 episodes of sustained ventricular tachycardia in the monitoring zone. This information was added to the results part of the manuscript.
3- Were the ECG parameters measured manually?
All the measured ECG parameters were made manually by a cardiologist blinded to the patient data.
4- The English needs to be edited. Several sentences are unclear. One example is the following: ‘All patients 136 were previously doing an ACEI or ARB and a beta-blocker and 94.3% were doing an MRA.’
As suggested, the English was reviewed. Regarding the sentence highlighted by the Reviewer, it was changed to “All patients were previously doing an ACEI or ARB and a beta-blocker. Furthermore, 94.3% were doing an MRA.”
5 - Many sections in the discussion are identical to the sections in the Introduction (see manuscript page 6, lines 184-199). The authors should refrain from copy-pasting the same sentences, and should rather focus on the implications of their findings in the context of previous reports. The whole discussion should be restructured, rewritten and shortened.
As suggested, the discussion section was restructured and shortened. Several changes were made in the revised manuscript.
6 - The authors shall also discuss the significant changes observed in the ‘SV1 + RV5/6‘. Do the authors that this is due to a reverse remodeling of the LV within 6 months?
A paragraph was added in the manuscript to discuss the changes observed in the SV1+RV5/6.
Despite we found no reduction in interventricular septum dimensions after 6 months of LCZ696 therapy (9.6 ± 1.7mm at baseline vs 9.9 ± 1.9mm at 6 months, p=0.280), previous studies had already shown that an increased LV mass will not change the QRS amplitude unless sufficient concurrent chamber dilatation was present (1, 2). Since our results revealed signs of reverse remodeling after 6 months of LCZ696 therapy, including a reduction of LV dimensions and an improvement in LV ejection fraction, without an increase in septal thickness, a reduction in the SV1+RV5/6 could be expected.
We will be very pleased to clarify any doubt left.
Yours sincerely,
António Valentim Gonçalves, M.D.
REFERENCES
1. Antman EM, Green LH, Grossman W. Physiologic determinants of the electrocardiographic diagnosis of left ventricular hypertrophy. Circulation. 1979;60(2):386-96.
2. Sundstrom J, Lind L, Andren B, Lithell H. Left ventricular geometry and function are related to electrocardiographic characteristics and diagnoses. Clinical physiology. 1998;18(5):463-70.

Reviewer 2 Report
In this manuscript Goncalves and colleagues assess the impact of sacubitril/valsartan on arrhythmic burden and assess the relation with electrical and mechanical remodeling. The authors assess this in a prospective single center cohort that is stingingly well treated with guideline directed therapies (both pharmacotherapy and device based interventions). Authors show that initiation of sacubitril/valsartan is associated with mechanical and electrical remodeling, however. I have several comments; Perhaps an analysis on the association between arrhythmic burden (most patients had an ICD) and electrical and mechanical remodeling could further strengthen the manuscript.
1. NT proBNP is not normally distributed in hardly any HF population, please report as median and interquartile ranges. Additionally, NTproBNP is relatively low for a patient population that is predominantly in NYHA class III or IV. Do authors have an explanation?
2. Authors report on page 4 that 6 patients before and 2patients after initiation of sacubitril/valsartan had an an-rrhtyhmic event. DO author have more data from the periodic device analysis, as almost all patients had an ICD (eg nsVT burden, mean duration nsVT, PVC burden, appropriate therapies, VTs entering monitoring zones ect?)
3. Following sacubitril/valsartan two patients suffered SCD, the patients that suffered SCD did they have worsening in mechanical dispersion or electrical dispersion?
4. Do authors have data on nsVT burden during follow-up from the device analysis? If so, did the patients that have worsening of mechanical and electrical dispersion also have more non sustained VT? Authors could use the burden of nsVT as a marker for more severe ventricular arrhythmias.
5. In the discussion authors should at and discuss the manuscript from Martens et al (Clin Res Cardiol. 2019 Feb 20. doi: 10.1007/s00392-019-01440-y.) that also shows a reduction in arrhythmic burden, which is linked to cardiac reverse remodeling.
Author Response
To: Reviewer 2
1- Perhaps an analysis on the association between arrhythmic burden (most patients had an ICD) and electrical and mechanical remodeling could further strengthen the manuscript.
5 - Do authors have data on nsVT burden during follow-up from the device analysis? If so, did the patients that have worsening of mechanical and electrical dispersion also have more non sustained VT? Authors could use the burden of nsVT as a marker for more severe ventricular arrhythmias.
We agree that the association between arrhythmic burden, as assessed by ICD, and electrical and mechanical remodelling would further strengthen the manuscript. However, we were positive surprised by the low number of arrhythmic burden in the 35 patients of our trial, since only 2 patients had sustained ventricular arrhythmias in the ICD monitoring during the 6 months of follow-up. Furthermore, only 3 patients had at least 1 episode of non-sustained ventricular tachycardia (defined as ≥ 4 consecutive beats with a monitor zone rate cut-off of 140bpm) in the ICD monitoring, all of them with slight numerical improvement in left ventricular ejection fraction and global longitudinal strain, but no reduction in QTc interval and mechanical dispersion. In all other 25 patients (5 patients did not have an ICD system), there were no episodes of sustained or non-sustained ventricular tachycardia recorded.
A reduction in sudden cardiac death (3) arrhythmic burden as assessed by ICD monitoring following LCZ696 therapy has already been reported (4, 5) and was not the aim of this study. Instead, we aimed at analyzing the ECG and echocardiographic (as assessed by mechanical dispersion) changes with LCZ696 therapy, which could potentially provide new insights to the antiarrhythmic mechanisms associated with this therapy. Nevertheless, the information on the arrhythmic burden as assessed by ICD monitoring before and after LCZ696 therapy was added to the manuscript as it adds to the consistency of our results.
2- NT proBNP is not normally distributed in hardly any HF population, please report as median and interquartile ranges. Additionally, NTproBNP is relatively low for a patient population that is predominantly in NYHA class III or IV. Do authors have an explanation?
The baseline value provided was BNP and not NTproBNP. The median baseline value of NT-proBNP in our population was 1873pg/ml.
We initially presented mean ± (SD) values for BNP because normal distribution was not excluded by the Kolmogorov-Smirnov test (p 0.252). However, we agree with the Reviewer that BNP should be expressed by median (IQR), considering the outliers and the distribution of this variable in previous registries and trials. We have modified the manuscript accordingly.
The median baseline value of BNP in the PARADIGM-HF trial was 202 pg/ml (6), which is numerically lower than the median BNP value (314 pg/ml) in our sample, although we recognize that the majority of the patients (75.9%) were in NYHA class ≤ II in the aforementioned trial. The BNP values in our sample may reflect a stabilized phase of HF and an effective decongestion therapy, despite the presence of advanced symptoms, possibly related to low cardiac output.
3- Authors report on page 4 that 6 patients before and 2 patients after initiation of sacubitril/valsartan had an an-arrhtyhmic event. DO author have more data from the periodic device analysis, as almost all patients had an ICD (eg nsVT burden, mean duration nsVT, PVC burden, appropriate therapies, VTs entering monitoring zones ect?)
As previous mentioned in this response, in these 2 patients, ICD monitoring revealed a reduction in the number of ventricular arrhythmias. One patient had 5 episodes of ventricular tachycardia in the monitoring zone plus 183 episodes of non-sustained ventricular tachycardia in the 6 months previous to LCZ696 therapy. In the same LCZ696 therapy period, no maintained ventricular tachycardia episodes were recorded, and there was a reduction to 70 episodes of non-sustained ventricular tachycardia. In the other patient, there was a reduction from 5 episodes of ventricular tachycardia treated appropriately with burst pacing to only 2 episodes of ventricular tachycardia in the monitoring zone. This information was added to the results section of the manuscript.
4- Following sacubitril/valsartan two patients suffered SCD, the patients that suffered SCD did they have worsening in mechanical dispersion or electrical dispersion?
Regarding the 2 deaths in our population, only 1 patient had sudden cardiac death, considering that the other patient died due to intracranial hemorrhage following head trauma (not due to syncope).
The patient with sudden cardiac death had no ICD (baseline LV ejection fraction of 38%). Baseline value of mechanical dispersion was 102 msec and QTc interval duration of 465 msec. Since he did not complete the six-month follow-up with LCZ696 therapy, we do not know if the mechanical dispersion value changed. However, an ECG was recorded at four months of LCZ696 therapy (one month before death), showing a QTc interval of 462 msec.
6 - In the discussion authors should at and discuss the manuscript from Martens et al (Clin Res Cardiol. 2019 Feb 20. doi: 10.1007/s00392-019-01440-y.) that also shows a reduction in arrhythmic burden, which is linked to cardiac reverse remodeling.
As suggested, the main information of this article was added to the introduction section:
“This antiarrhythmic effect was confirmed in two other trials, in which LCZ696 therapy was associated with a significant reduction in episodes of non-sustained and sustained ventricular tachycardia, appropriate ICD shocks, premature ventricular contractions and consequently, an increase in biventricular pacing percentage (4, 5).”
and to the discussion part of the manuscript:
“One previous retrospective analysis of real-world patients revealed that a higher degree of reverse remodelling was associated with a lower burden of ventricular arrhythmias, as assessed by ICD monitoring (4).”
We will be very pleased to clarify any doubt left.
Yours sincerely,
António Valentim Gonçalves, M.D.
REFERENCES
1. Antman EM, Green LH, Grossman W. Physiologic determinants of the electrocardiographic diagnosis of left ventricular hypertrophy. Circulation. 1979;60(2):386-96.
2. Sundstrom J, Lind L, Andren B, Lithell H. Left ventricular geometry and function are related to electrocardiographic characteristics and diagnoses. Clinical physiology. 1998;18(5):463-70.
3. Desai AS, McMurray JJ, Packer M, Swedberg K, Rouleau JL, Chen F, et al. Effect of the angiotensin-receptor-neprilysin inhibitor LCZ696 compared with enalapril on mode of death in heart failure patients. European heart journal. 2015;36(30):1990-7.
4. Martens P, Nuyens D, Rivero-Ayerza M, Van Herendael H, Vercammen J, Ceyssens W, et al. Sacubitril/valsartan reduces ventricular arrhythmias in parallel with left ventricular reverse remodeling in heart failure with reduced ejection fraction. Clinical research in cardiology : official journal of the German Cardiac Society. 2019.
5. de Diego C, Gonzalez-Torres L, Nunez JM, Centurion Inda R, Martin-Langerwerf DA, Sangio AD, et al. Effects of angiotensin-neprilysin inhibition compared to angiotensin inhibition on ventricular arrhythmias in reduced ejection fraction patients under continuous remote monitoring of implantable defibrillator devices. Heart rhythm. 2018;15(3):395-402.
6. Myhre PL, Vaduganathan M, Claggett B, Packer M, Desai AS, Rouleau JL, et al. B-Type Natriuretic Peptide During Treatment With Sacubitril/Valsartan: The PARADIGM-HF Trial. Journal of the American College of Cardiology. 2019;73(11):1264-72.

Round 2
Reviewer 1 Report
I thank the authors for addressing my comments. The revised manuscript has significantly improved, yet I find the language quite confusing at many points.. The English still needs editing, because in a number of places, particularly the newly written parts of the discussion, the words and phrases used are unclear and do not reflect what the authors were likely to express (e.g. lines 201-202: ‘… patients were previously …’, line 272, ‘… since unlikely BB …’, etc).
doing